# A Pilot Study Conducting Online Think Aloud Qualitative Method during Social Distancing: Benefits and Challenges

**DOI:** 10.3390/healthcare10091700

**Published:** 2022-09-05

**Authors:** Asim Alhejaili, Heather Wharrad, Richard Windle

**Affiliations:** 1School of Health Sciences, University of Nottingham, Nottingham NG7 2HA, UK; 2College of Nursing, Taibah University, Medina 42353, Saudi Arabia

**Keywords:** Think Aloud, online method, data collection, nursing

## Abstract

COVID-19 social distancing restrictions provided unprecedented insights into online research methodologies and approaches for both participants and researchers. Field research traditionally conducted face-to-face had to be transferred online, highlighting the great strides made in communication technologies (particularly live video streaming) over the last two decades for online qualitative research. However, dedicated research on these phenomena is tentative, including with regard to specific methods such as Think Aloud. This paper contributes to literature on online Think Aloud in qualitative research, evaluating new insights on its adoption online. It draws on findings from an online piloting study of Think Aloud tasks to explore the implications of using real-time internet video calls via SoIP applications by MS Teams. To assess the online Think Aloud process, this review called upon some of the comments made by participants during the semi-structured interview or comments made during the Think Aloud process, when they were relevant to the online process itself. It focuses on different dimensions of benefits, rapport in the session’s encounter, challenges, and ethical concerns. Overall, the findings indicate that online Think Aloud sessions cannot completely replace in-person sessions for some particular and highly in-depth research areas, but they can greatly facilitate qualitative data collection in most conventional contexts. It is necessary to carry out further studies exploring the use of this and other online approaches and instructions.

## 1. Introduction

Conducting research during the COVID-19 pandemic has provided unprecedented insights into the potential and pitfalls of online research methodologies and approaches [1]. Enforced social distancing practices meant field research originally being conducted face-to-face had to be transferred online during COVID-19 [2]. A considerable amount of literature has been published on the benefits and limitations of online methods in general, which offer obvious expedient and practical advantages in terms of ease of recruitment and ability to recruit from various locations [3,4]. Collecting data online obliterates traditional time and geographical barriers and financial constraints to researchers, all of which adversely affect onsite data collection, but it has traditionally been viewed as sacrificing quality in the interests of expediency. For instance, when conducting quantitative data about personal and/or emotional subjects, the traditional face-to-face, in-person interview has been assumed to be preferable due to the trust and rapport fostered between interviewers and interviewees, which could not be emulated by traditional distance interview methods such as telephone or email interviews [5,6].

However, technological developments over the last few decades, particularly the improvements relating to imaging and high-speed internet, have revolutionized the capabilities of online communication, present new and exciting opportunities for research work [7]. Limitations relating to the potential lack of depth due to difficulties in probing and facilitating discussion online [8,9], and the inability of the researcher to formulate impressions and pick up on non-verbal cues [10], seem increasingly luddite in the face of crisp and clear modern video conferencing capabilities available in most developed-world research contexts. However, in contexts where advanced internet capabilities are not available, particularly for vulnerable and disadvantaged groups, the traditional limitations of distance interview methods remain pertinent. Furthermore, modern video-conferencing methods may exclude those with little or no access or know-how to work with technology [11]. In this paper, the transfer of the Think Aloud (TA) approach online is outlined using a pilot study in nursing, to see if the main study can proceed with or without modifications. Before talking about how the TA approach transfers online, the next section explains the provenance of TA and how it traditionally operates in face-to-face contexts.

### 1.1. Think Aloud (TA)

TA as a qualitative data collection method has been widely utilized in cognitive psychology as a means of gathering verbalizations regarding productive thinking and as a means of understanding the development of thought in individuals [12]. Three methods identified to produce verbal reports: concurrent protocol (in which participants verbalize their thoughts while performing a task or activity), retrospective protocol (in which comments describing the task are gathered after performing the task), and post-reflective protocol (where the participants are encouraged to give reasons for their actions) [13]. This approach relates to the mental process of the verbalized individual, especially the sequence of thinking processes or cognitive events between the predefined task or problem and the final outcome. Verbalizations (i.e., the eponymous “think aloud”) are a subgroup of the cognitive processes that result in the execution of an action or behavior [14].

TA is a research method in which participants speak aloud any words in their mind as they complete a task [15]. A review of the literature has shown that TA research methods have a sound theoretical basis and provide a valid source of data about the psychological mechanisms and knowledge structures underlying human problem-solving activities with respect to specific tasks, including problem solving, reading, composing, second language learning research, counselling, business, and human–computer interactions, etc. [16,17,18]. TA is commonly used to gather verbalizations with high-quality user feedback [19]. The use of the TA method to observe the participants provides an accurate reflection of their behavior [14,20].

TA data can be collected from a variety of sources, including direct observation of participants who share details of what they are trying to do, as opposed to simply telling the observer what they think they want to hear (which creates social desirability bias) [21]. The verbal response is recorded with an audio/video recorder or by taking observation notes as participants work through the Think Aloud Protocol (TAP) [22]. These have to be analyzed and examined systematically and in depth in order to generate data about each participant’s behavior and their approach to cognitive reasoning, as demonstrated while they act [23]. Raw TAP data needs further processing and interpretation to provide deep insights on the way in which individuals perform activities [17].

### 1.2. Moving to Online Methods

The literature search undertaken by the researchers identified no reports of modifying the TA methodology for online delivery, and the nature of online TA remains unclear. To our knowledge, no one has investigated “online TA”, where participants are seated and perform TA tasks remotely within a session over Internet Protocol (SoIP) network. Therefore, this paper attempts to address this by providing a critical reflection on the implications of online TA. With this study, we hope to help contribute to this underdeveloped area in the online qualitative methodology literature. It is worth noting that the points we raise in this article are of a conceptual nature, focusing on developing recommendations for the use of online TA as an observation tool in qualitative research.

In this paper, we present clinical statements task to conduct online TA via SoIP applications by Microsoft Teams (MTs). The participant connects via video conferencing tool to do online clinical statements task related to nursing work. The goal of our research is to understand the necessary of exploring best approach to conduct actual online TA involving nursing students. The two core objectives guiding our research are as follows:To illustrate how TA could be conducted online and evaluate transferring the approach online;To consider the benefits and challenges of remote data collection in a TA study.

## 2. Materials and Methods

### 2.1. Study Design

To fulfil its aim, the current pilot study utilized a clinical statements task to communicate participants’ intent to seek information through different online resources, thus engaging the participants to contribute to concurrent TA in a remote setting. Participants were given instructions and a brief description of the TA technique. For each clinical statement, a PowerPoint slide popped up to show the new clinical statements with instruction to guide them through a task. The mean time to complete the tasks was 43 (+/− SD) minutes for the nine clinical statements. The nine statements represented a typical nursing care encounter, with decision-making focused on a request for formulating a clinical decision to a patient. Then, the participants asked to seek and obtain evidence-based information related to the clinical statement.

This task was followed by semi-structured interviews conducted to explore benefits and challenges during the online TA sessions. The data obtained through the interviews were then analyzed thematically using Braun and Clarke [24] guidelines to explore the emergent factors that related to online TA from participants’ perspectives [25]. This method allowed us to concentrate on recruiting participants, developing rapport with them, and honoring ethical requirements in order to gain a better understanding of the specific needs within each of the identified themes. Our current analysis focused on the benefits and challenges of online TA. It included an analysis of findings from the semi-structured interviews along with some online TA troubleshooting. The data was digitally recorded with the consent of participants, transcribed, and imported for analysis. Data collection and analysis was undertaken by the principle researcher, and verified and reviewed within the research group.

### 2.2. Participant Recruitment

Seven participants were recruited to conduct the pilot test. These participants were subject to the same inclusion and exclusion criteria as those who took part in the main study (as described below), but their results were not included for analysis in the latter. Various studies have assessed the efficacy of various pilot study sample sizes. While there is no universal blueprint, the sample is inevitably relatively small, particularly for qualitative research [26,27,28]. One rule of thumb is that the pilot study sample should be 10% of the final study sample size [27]. The main study aims to recruit 14 participants for the actual TA, which compares favorably with a similar previous empirical study which used a convenience sample of five participants [29]. This study started with TA sessions followed by semi-structured interviews to evaluate the usability of a system.

The sample in the pilot research must be similar to the sample in the main study (in order to be effective in piloting the research instruments as a testing phase) [26]. Sampling is based on the relevance of the participants to answer the research question. Inclusion criteria were calibrated to recruiting a sample of nursing students in possession of a laptop with camera and audio input and share screen capability. They completed all the educational requirements at Taibah University. At the time of recruitment, an invitation letter was sent to 30 nursing interns who had begun the program. Four participants (P1–P4) were nursing intern students at an internship program at Taibah University (Saudi Arabia). Furthermore, two PhD students (P5 and P6) and one experienced TA researcher (P7) volunteered to complete the same online task to experience doing the online sessions.

They all then participated in semi-structured interviews in order to explore their experiences of taking part in a TA session online approach. Verbal data were gathered, documented, and transcribed, and was then thematically analyzed [24]. This entails a constant cross-checking process between the data set, coded extracts, and generated data [30]. The developed themes were checked repeatedly to determine whether it was appropriate to split or merge themes [25]. Feedback and thoughts about the online TA and suggestions for improvement were gathered from the participants during the semi-structured interviews. These codes were then shared with and critiqued by the research team.

### 2.3. Ethics

Ethics is central to data collection methods in every piece of research [31,32]. Ethical concerns include those pertaining to intellectual property, informed consent, right to withdraw, unintended deception, accuracy of portrayal, confidentiality, and financial gain [33]. With the advent of online research, the above concerns are still valid, but less easy to define. In particular, the online world raises ethical issues around access to data and techniques for the protection of privacy and confidentiality [34].

The study was carried out after receiving ethical approval from the Research Ethics Committee of the Faculty of Medicine and Health Sciences at Nottingham University, and the Faculty of Nursing at Taibah University. Nursing and PhD students read the participant information sheet (PIS) and completed the consent form online. Participants were informed that their TA and semi-structured interview sessions would be recorded with their permission, and that they could stop the recording at any time or withdraw from the research. They were given the opportunity to choose the day and time of their sessions and interviews.

### 2.4. Methodological Modifications to the Traditional TA

This study was part of a PhD project at the University of Nottingham. The PhD study aimed to explore the information seeking behavior (ISB) factors and strategies that contribute to obtain evidence-based information within healthcare among nursing intern students (NISs). Due to COVID-19 social distancing practices, field research hitherto normatively conducted face-to-face had to be transferred online (otherwise it had to be suspended or cancelled altogether). Implementing TA online is a challenging task, as it requires simulating all tasks that are performed in the face-to-face settings, therefore, MS Teams and PowerPoint were used to remotely replace onsite activities. Interviews were employed to evaluate and gain important new insights of conducting TA method, adapted, and transferred online. Participants were instructed to TA to ensure that all the activities that they engaged in during the information seeking process were captured on the shared screen and audio recording to represent an actual TA. Figure 1 shows the instructions presented to the participants before starting the online TA session.

In the TA, participants were asked in the presence of the researcher to vocalize their thoughts, actions, expectations and articulate any confusion or concerns that arose. For the different information-seeking exercises, they had access to a range of web resources or they could navigate health information freely, such as Googling to obtain resources. Online sessions were recorded using Microsoft Teams sharing and recording options. The TA session was conducted using a laptop with audio input, and they asked to share the screen. Figure 2 shows an example clinical statement and how a participant interacted during the thinking aloud online session.

### 2.5. Evaluation of the Online TA Method

It is important to consider the perspectives of both participants and researchers in terms of the process of conducting and contributing in an online TA study, thus direct observation of TA was used. Direct observation is a popular method to explore participants’ understanding of online sessions. This was followed by reflective semi-structured interviews, to gain a deeper understanding of participants’ experiences of the online TA study (Table 1).

## 3. Results

The results of this study indicate several factors perceived by participants as beneficial and challenging regarding online TA. The interview questions were used to organize the presentation of the data. Overall, there was agreement among researchers and participants that online TA was a useful method for conducting qualitative interviews. Participants frequently reported the following points, reflecting impersonal, technical, and ethical considerations: (1) recruitment, (2) rapport, and (3) ethical concerns. Similar studies that looked at online methods to collect qualitative data also identified the salience of these categories [2,9,35]. Quotes provided in this section are illustrative excerpts from interviews of themes that were common across multiple participants’ accounts (the analysis of the interview transcripts formed the basis of the thematic analysis presented below).

### 3.1. Recruitment

The majority of students agreed that using Microsoft Teams helped greatly widen the range of our recruitment without being physically present, thus incorporating wider experiences in the research by allowing the researcher to reach many different participants without geographical limitations. A couple of students mentioned that it was important to consider the clinical workload and assigned an appropriate time to encourage participation. P5 said this method allowed respondents more flexibility with regard to participating, since he had lack of time due to the workload in clinical practice. He thought that doing the TA session online provided more convenience in terms of time and space, *“Yes, because I have a lot to do in clinical practice regarding nursing responsibilities, so I could now find a suitable time out of my duty shift to participate*”. Additionally, researchers’ assistance and instruction were found to be very useful, and some students found written instructions more helpful than oral instructions. As P3 noted, “*I think written instructions should be fine, because it was easy to go back and read them*”.

Participants’ concerns were expressed about technological expertise of familiarity with using MS Teams. Two participants revealed that the using MS Teams was complicated, which they mainly attributed to a lack of familiarity. They did not know how to use MS teams, which meant that those with little technological expertise needed to be trained. One asked to use another platform instead of MS Teams. P2 said “*Because it was my first time and I had no idea how to use Microsoft Teams. Can I use Skype for the session instead of Microsoft Teams?*”. Moreover, P3 asked the researcher to show him how to use MS Teams as follows:


*P3: I do not know how to use Microsoft Teams in this task. Could you teach me?*



*Researcher: I will share my screen and explain to you how to use it.*


Another common obstacle to using online TA was the ability to access to a computer with the necessary software. Participants were not able to choose to have the TA sessions face-to-face, which meant that those with no laptop might be excluded. P6 was not able to volunteer for an online session at the chosen time and needed to reschedule the TA session:


*Researcher: Which device do you use to enter Teams?*



*P6: Hi! I am using my smartphone?*



*Researcher: Sorry, can you use a laptop to enter? Because it is necessary to participate in this study, due to the nature of the Think Aloud session.*



*P6: I do not have a laptop right now.*



*Researcher: Sorry, I have to rearrange this session for another time, thank you for your willingness to participate.*


### 3.2. Rapport

Participants mentioned a number of advantages of the online TA that enhance rapport. Participants who were not comfortable with being physically observed could be anticipated to experience stress. However, most participants reported that online TA has some flexibility. This was supported by P3, who mentioned that doing the TA task online could be more relaxed and less intimidating than face-to-face activities in the same physical space which maintain rapport. He said that “*I think because of physical distance, I feel more relaxed to do the task without being observed and evaluated in the physical presence of a researcher*”.

Correspondingly, P7 thought that the online session helped to avoid embarrassment brought about by presence of a researcher. He expressed a preference for online tasks, arguing that he felt more comfortable not being able to observe his facial reactions or the researcher expressions. He stated that, “*Because I can’t see your facial expressions. I feel more comfortable to interact in the session more than speaking to you face-to-face*”.

There were some occasions when the researcher needed to say prompting words during the sessions, whenever the participants became silent to resume their actions and thoughts, which allowed us to obtain better data. For example, in one session there were some prolonged pauses (silence), during which participants were induced to speak to ensure the maintenance of rapport:


*Participant: [Silent for 40 s]*



*Researcher: Keep on talking…*



*Participant: I will thanks.*

*[P2]*


There was no problem with regards to online sessions interrupted by the loss of connection. In one session the internet video call was established as the audio and video clarity were checked, then the interaction was lost. I then sought clarification of the connection quality, instead of continuing with the exchange of conversations. After that I resumed the conversation to sustain rapport:


*Participant: [Call disrupted for 20 s]*



*Researcher: Hi! I cannot hear you?*



*Researcher: How are you?! [Connection back]*



*Participant: I’m good thanks.*



*Researcher: Can you complete sharing screen and resume again?*



*Participant: I can... yes, It’s fine.*

*[P4]*


### 3.3. Ethical Concerns

For the purpose of carrying out online TA sessions in this research, the researcher followed standard ethical procedures to address all ethical concerns. Therefore, the volunteers had a chance to pre-read the consent form, and they were pre-warned and asked if the interviews could be recorded. They were informed that the recording of the sessions could be stopped at any time on request, and that they could withdraw at any time from the research; they were given the opportunity to choose the location, day, and time of their session. With the advent of online research, the privacy and confidentiality concerns are still valid. With conducting TA sessions through Microsoft Teams, there are some additional ethical considerations to take into account, namely the issues created by the fact that the interaction is mediated through the use of technology (which is owned by third parties), the verification of participant’s identity; and issues raised by the interview environment and the nature of recording this. P2 in response to difficult of participation mention that it was important for both the researcher and the platform to protect all participants’ identity information. He stated that, “*It might be an issue in terms of privacy and confidentiality with online sessions, particularly for me as a student, in terms of evaluating my nursing performance. I might be concerned if this session was shared with my clinical instructors*”.

To start the session, participants needed to open a new window in the search engine. The researcher noted that this might show some previous webpages sought by the participants, and thus asked the participants to close all the windows and opened one window with erasing their previous history, in order to maintain their privacy. A couple of participants felt uncomfortable about the potential to show their search history unintentionally *(“It would be better to delete previous search history”).*

## 4. Discussion

Despite the challenges described above, online TA was overwhelmingly perceived to be beneficial by the participants. One main reason given was that the online approach provided the participants with flexibility in terms of not having to be in a particular physical location to attend. Using an online approach, the researcher was able to transcend geographical boundaries, nullifying distances and eliminating the need to visit an agreed location for interview [2,36,37]. Also, when using MS Teams and other similar technologies, TA session can easily be conducted from the comfort of one’s home, eliminating not only the need to travel, but also the need to find a venue to do the TA sessions [38,39]. Logistical issues were also eliminated in this study with regards to access to certain spaces such as a classroom, meeting room, and hospital areas. However, participants may have felt uncomfortable being recorded or filmed in their own homes.

Ideally, researchers need to decide on the type and level of difficulty of the research processes, and the degree of prompting which is appropriate to meet the demands of a TA task [40,41]. With the use of online methods, materials used in the TA task were minimized by the researcher, to facilitate conducting the fieldwork in a more flexible way, built around the needs of participants. However, two participants needed a tutorial session on how to use MS Teams, which could be linked to the cognitive overload of using the technology in itself, in addition to the focus on the content of the activity [42]. It would be preferable to send pre-prepared tutorial videos before sessions to avoid any anxieties or delay during the real tasks. Potential participants could be deterred and even refuse to participate due to a lack of familiarity with using MS Teams. Another obstacle to conducting online TA was the issue of access to a computer with the necessary software and the ability to use it during the session (as explained above, P6 had expected to be able to conduct the task using a smartphone and did not have a laptop available). Therefore, an important inclusion criterion for participants is to be able to use a laptop with audio input and the capability to share screen for the duration of the task.

Previous studies mostly defined rapport as the degree of comfort in the interactions between the researcher and participants [43,44]. For participant observation, rapport refers to the quality of the relationships that the researcher makes at the field site [44,45]. One of the most important benefits of online TA was that participants felt more comfortable, which facilitated their task execution. A study of participants in a hospital setting were not able to speak openly, which undermined the richness of the data; the subsequent use of complementary telephone interviews produced much better data [46]. This finding was also reported by other studies [47,48]. Despite the reported benefits, the necessity of access to high-speed Internet, familiarity with online communication, and having digital literacy should be mentioned among the inclusion and exclusion criteria. Another contentious issue around SoIP technologies and their limitations is the building of rapport. For example, three of the seven interview participants indicated that they had difficulties with online communication. The absence of face-to-face interaction in online settings presents a difficult to interactions with participants [43]. Thus, there is a need to work with participants to overcome the difficulty of communication in an online environment.

In this method, there are several concerns with regards technical difficulties or loss of internet connection that might affect rapport. Similarly, digital issues can lead to a loss of consistency [49]. For instance, if the internet connection is lost during performing a TA task or an activity, it might generate an unexpected interruption and rupture of the TA session that it is subsequently hard to move forward from (after resuming the connection) [50]. In the context of the TA sessions, the researcher found that there was no problem with regards to rapport: even on those rare occasions when the call was interrupted by the loss of connection or screen freeze, there was no problem resuming the conversation. This is not a new challenge for online methods in general. Numerous researchers have studied the absence of connection in online methods [5,10,35]. To avoid a sense of ambiguity, the researcher should provide instructions to follow at the beginning of a project include expecting a soon reply by the research to resume the session or sending e-mail. These strategies can assist to let participants know about these challenges and help them to overcome the difficulties of online connection.

One of the most significant current discussions in online methods involves challenges to the traditional axiom that the rapport and richness of interpersonal interactions may be lost when using online methods, which is an increasingly archaic presumption when modern internet and communication technologies are available [51,52]. Although often overlooked in formal processes, rapport is essential to ethical practice, particularly in terms of building a research relationship founded on respect. While it might initially seem that it is harder to offer the same level of rapport via online tools such as MS Teams compared to offline face-to-face contexts, the reduced non-verbal communication cues in online TA compared to face-to-face sessions or on-site observational studies that might have limited rapport were not evident in this pilot study. Overall, this study has raised important questions about whether online or face-to-face TA sessions are better to build rapport. With conducting TA sessions through MS Teams, there are some additional ethical considerations to take into account, mentioned by the participants, which pertain to the interaction being mediated through the use of technology (which is owned by third parties), the verification of participant’s identity, and issues raised by the interview environment and the nature of recording. After data collection, data was transcribed and stored in password-protected computer files accessible only to the authors.

The COVID-19 pandemic required us to think creatively about the ways we conducted research to ensure that it was still robust, ethical, and safe, whilst being carried out at a distance. It is true to say that we have learnt a lot about the fundamentals of research and the affordances of new technology within this field. The TA process is a good example. TA is based on a social interaction between the participant and the researcher, as the researcher observes and questions the participant’s interaction with a particular artefact or process. Whilst, we might have considered physical proximity to be an essential part of this, we have learnt that this technique can function very well at a distance using new social technology. Moreover, this understanding opens up very real and exciting opportunities for the use of the technique, enabling flexibility in terms of location, as participants to be located anywhere globally as long as they have access to the internet and the factor being investigated, flexibility of time, meaning research can more easily be managed around clinical practice, for example, flexibility of environment, meaning that the TA process can be brought more easily into clinical environments, for example and flexibility of participation meaning that many more people can participate in such studies.

## 5. Conclusions

The COVID-19 pandemic has dramatically revolutionized the way research can be carried out and created a need to continue research activities in diverse fields while keeping research field safe and academically robust. Researchers have been compelled to find innovative solutions in order to conduct research, including TA sessions, as face-to-face sessions were no longer possible during periods when social distancing measures were applied. This paper has discussed the advantages as well as limitations of TA online sessions. According to the findings of this study, the participants were often less worried about time because they were already at home and in a comfortable environment. Hence, they tended to interact more in the online session. Also, other advantages were reported, such as issues related to participants’ safety, the ability to reach out to participants at a convenient time, and greater convenience for both participants and researchers during the TA sessions, which enhanced participant recruitment. Overall, while online sessions cannot completely replace in-person sessions, especially for certain contexts that really require personal presence in the same time and space, the irrefutable and compelling advantages of online TA offer an unprecedented opportunity to gather more voluminous and comprehensive data pertinent to answer research questions. Further studies are need using the same online approach and instructions to assay the relative quality and utility of such strategies to drive academic inquiry in future. We conclude by highlighting the main study will be feasible based on the pilot study.

## Figures and Tables

**Figure 1 healthcare-10-01700-f001:**
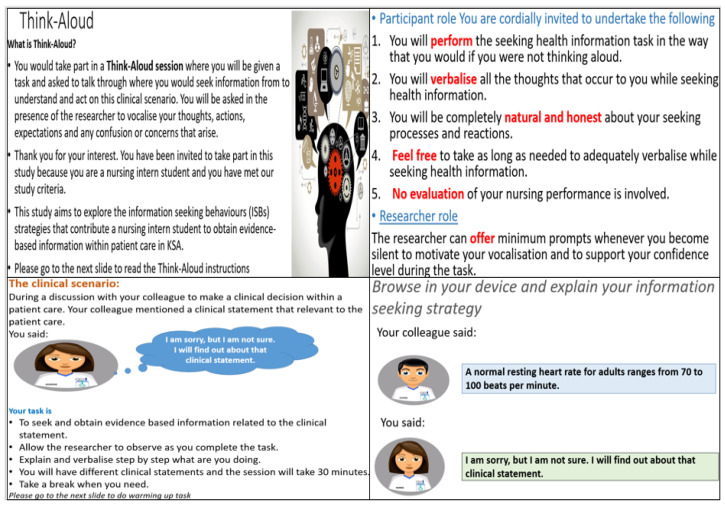
TA instructions.

**Figure 2 healthcare-10-01700-f002:**
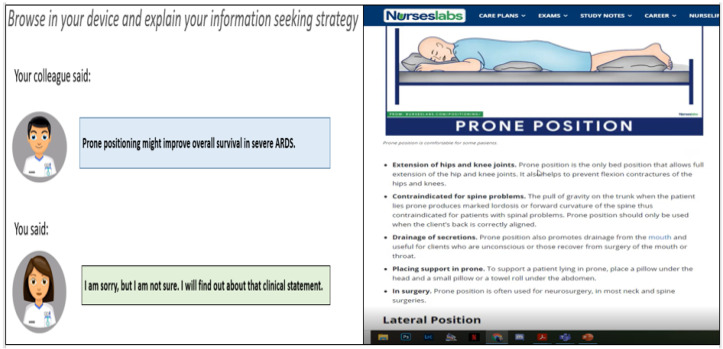
TA session.

**Table 1 healthcare-10-01700-t001:** Semi-structured interview questions.

How easy was it to participate in the online TA session?
How difficult was it to participate in the online TA session?
Do you think that doing a TA session online encouraged you to participate in the research?
Do you have any general comments about the online session?

## Data Availability

Data are contained within the article.

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
