# Peer review of "A Pilot Study Conducting Online Think Aloud Qualitative Method during Social Distancing: Benefits and Challenges"

_healthcare, 2022, doi:10.3390/healthcare10091700_

Round 1

Reviewer 1 Report

Revision of the article:

Conducting Online Think-Aloud as a qualitative method during the COVID-19 pandemic: Benefits and Challenges

This article aims to illustrate how Think Aloud could be conducted online and evaluate transferring the approach online, as well as to consider the benefits and challenges of remote data collection in a Think Aloud study.

The article has many areas for improvement prior to assessing its possible publication:

- At the end of the introduction section, there is an anticipate assessment of the Online Think-Aloud as a qualitative method that should go in the conclusions section.

- The introduction section needs a more extensive conceptual framework to justify the objectives of this study.

Figure 1 does not display well.

- The methods section needs more information to be able to better understand, and where appropriate, to be able to reproduce the present study. It is necessary to expand the information on the design and methodology of the study: how the participants were selected, what type of methodological approach was used, how the data were analyzed (if any computer program was used...) etc.

- It makes no sense to present table 2 with so little information, since it can be done in text mode.

- The writing of the results is very poor and shows little scientific rigor. The results are too poor to publish them in a scientific article.

- In the discussion section appears testimonials that should be included in the results section.

- It is necessary to rework the results section and the discussion section. Both sections are mixed.

- References are not well referenced according to the regulations of this journal and should be reviewed. You can check the recommendations in: https://www.mdpi.com/journal/healthcare/instructions

 In order to be publishable, the article needs a major improvement.

 Kind regards

Author Response

Please see the attachment, thank you!

Reviewer 2 Report

From a first look, the theme of the study and conducting this research during the COVID-19 pandemic seems very important. Such an approach has little coverage in the international literature. However, the way the study has been presented with significant missing points makes it problematic. Since, it is part of a Ph.D thesis, one would expect that the missing points would not appear in this paper. More specifically, the methodology is incomplete. The subjects’ background, the criteria for selecting them, the data collection procedures/steps, the instruments used for data collection, and the way the data are analysed are not clear at all.  It seems that a thematic analysis is employed but how, what themes were constructed, the validity, etc. are missing from the methodology section. It is guessed that a two-step analysis: an inductive and a deductive analysis, has been performed but not presented. It is a part of a Ph.D. thesis and all these missing points should be explored widely.   

Author Response

Please see the attachment, thank you!

Reviewer 3 Report

The paper reflects on the use of the think-aloud protocol in online studies, especially with lessons from the COVID-19 isolation period.  Reflections on how to apply the think-aloud method are valuable to a number of research areas.

In the introduction of the paper, the authors argue that they do not know any studies that examined the use of the think-aloud protocol in remote settings. In the field of Human-Computer Interaction, there are some examples of such studies, which I believe should be discussed in the paper:

Irlitti, A., Hoang, T., & Vetere, F. (2021). Surrogate-Aloud: A Human Surrogate Method for Remote Usability Evaluation and Ideation in Virtual Reality. In Extended Abstracts of the 2021 CHI Conference on Human Factors in Computing Systems (pp. 1-7).

Chalil Madathil, K., & Greenstein, J. S. (2011, May). Synchronous remote usability testing: a new approach facilitated by virtual worlds. In Proceedings of the SIGCHI Conference on Human Factors in Computing Systems (pp. 2225-2234).

Wood, R., Dixon, E., Elsayed-Ali, S., Shokeen, E., Lazar, A., & Lazar, J. (2021). Investigating Best Practices for Remote Summative Usability Testing with People with Mild to Moderate Dementia. ACM Transactions on Accessible Computing (TACCESS), 14(3), 1-26.

I recommend the authors revise Section 1.2 to make reading more fluid.

Figure 1 is very difficult to read.

Considering that the main focus of the paper was on the Think-Aloud protocol, I would expect a deeper discussion of definitions and concepts surrounding the method, such as (to name a few):

Jääskeläinen, R. (2010). Think-aloud protocol. Handbook of translation studies, 1, 371-374.

Fan, M., Lin, J., Chung, C., & Truong, K. N. (2019). Concurrent think-aloud verbalizations and usability problems. ACM Transactions on Computer-Human Interaction (TOCHI), 26(5), 1-35.

Obead Alhadreti and Pam Mayhew. 2018. Rethinking thinking aloud. In Proceedings of the 2018 CHI Conference on Human Factors in Computing Systems (CHI’18). 1--12.

K. Anders Ericsson and Herbert A. Simon. 1984. Protocol Analisys: Verbal Reports as Data. MIT Press.

Despite presenting a good level of detail about the questions regarding the think-aloud protocol, there was very little detail about the evaluation that was being conducted with the online think-aloud.

There was no mention of ethical approval by a Research Ethics Committee.

I found the organization of the paper somewhat confusing. I expected to see a deeper analysis of the interviews in the results, with a clearer organization that did not come only from the questions directly.  The lack of analysis also reflected on the fact that there was new data being added in the discussion. 

The paper also needs revision in its writing. Following I mention some examples of issues I have found.

Revise the use of articles: conducting Think-Aloud method

The first paragraph of the text did not describe TA as an acronym for Think Aloud

There are many statements that are difficult to read due to cumbersome wording, such as “one of which is it provides a unique…” on line 39.

Overall, I think the paper has good potential.  However, it needs substantial work to address related literature properly, a more precise conceptual discussion of the think-aloud protocol, and a better organization of the results.

Author Response

Please see the attachment, thank you!

Round 2

Reviewer 1 Report

View the attached file please.

Author Response

Please see the attachment, thank you!

Reviewer 3 Report

The authors have made important improvements to their paper.

The current version now has a good background with definitions of what the Think-Aloud Protocol is and its particularities. 

Despite not being comprehensive, the paper now includes a short review on related studies that used the Think-Aloud protocol in remote settings.

Data presentation in the results has been improved as well, with a better description of the categorization strategy.

Some figures still have poor legibility, and I suggest they be revised in a final version, should the paper be accepted.

Author Response

Please see the attachment, thank you!
